# HR-LCMS-Based Metabolite Profiling, Antioxidant, and Anticancer Properties of *Teucrium polium* L. Methanolic Extract: Computational and In Vitro Study

**DOI:** 10.3390/antiox9111089

**Published:** 2020-11-05

**Authors:** Emira Noumi, Mejdi Snoussi, El Hassane Anouar, Mousa Alreshidi, Vajid N. Veettil, Salem Elkahoui, Mohd Adnan, Mitesh Patel, Adel Kadri, Kaïss Aouadi, Vincenzo De Feo, Riadh Badraoui

**Affiliations:** 1Department of Biology, College of Science, University of Hail, P.O. Box 2440, Ha’il 2440, Saudi Arabia; eb.noumi@uoh.edu.sa (E.N.); mousa.algladi@gmail.com (M.A.); vajidnv@gmail.com (V.N.V.); salemelkahoui@gmail.com (S.E.); drmohdadnan@gmail.com (M.A.); riadh.badraoui@fmt.utm.tn (R.B.); 2Laboratory of Bioressources: Integrative Biology and Recovery, High Institute of Biotechnology-University of Monastir, Monastir 5000, Tunisia; 3Laboratory of Genetics, Biodiversity and Valorisation of Bioressources, High Institute of Biotechnology-University of Monastir, Monastir 5000, Tunisia; 4Department of Chemistry, College of Science and Humanities in Al-Kharj, Prince Sattam bin Abdulaziz University, Al-Kharj 11942, Saudi Arabia; anouarelhassane@yahoo.fr; 5Laboratory of Bioactive Substances, Center of Biotechnology of Borj Cedria, BP 901, Hammam lif 2050, Tunisia; 6Bapalal Vaidya Botanical Research Centre, Department of Biosciences, Veer Narmad South Gujarat University, Surat 395007, India; patelmeet15@gmail.com; 7Department of Chemistry, College of Science and Arts in Baljurashi, Albaha University, Albaha 65527, Saudi Arabia; lukadel@yahoo.fr; 8Department of Chemistry, Faculty of Science of Sfax, University of Sfax, BP 1117, Sfax 3000, Tunisia; 9Department of Chemistry, College of Science, Qassim University, Buraidah 51452, Saudi Arabia; kaiss_aouadi@hotmail.com; 10Laboratory of Heterocyclic Chemistry, Natural Products and Reactivity, Department of Chemistry, Faculty of Science of Monastir, University of Monastir, Monastir 5019, Tunisia; 11Department of Pharmacy, University of Salerno, Via Giovanni Paolo II, 132, Fisciano, 84084 Salerno, Italy; 12Section of Histology—Cytology, Medicine College of Tunis, Tunis El Manar University, Road Djebel Lakhdhar, La Rabta-Tunis 1007, Tunisia; 13Laboratory of Histo-Embryology and Cytogenetic, Medicine College of Sfax, Sfax University, Sfax 3029, Tunisia

**Keywords:** *Teucrium polium* L., HR-LCMS, phytochemistry, antioxidant, anticancer, Walker 256/B, MatLyLu, molecular docking

## Abstract

In this study, we investigate the phytochemical profile, anticancer, and antioxidant activities of *Teucrium polium* methanolic extract using both in vitro and in silico approaches. The results showed the identification of 29 phytochemical compounds belonging to 13 classes of compounds and 20 tripeptides using High Resolution-Liquid Chromatography Mass Spectrometry (HR-LCMS). 13R-hydroxy-9E,11Z octadecadienoic acid, dihydrosamidin, valtratum, and cepharantine were the main compounds identified. The tested extract showed promising antioxidant activities (ABTS-IC_50_ = 0.042 mg/mL; 1,1-diphenyl-2-picrylhydrazyl (DPPH)-IC_50_ = 0.087 mg/mL, β-carotene-IC_50_ = 0.101 mg/mL and FRAP-IC_50_ = 0.292 mg/mL). Using both malignant Walker 256/B and MatLyLu cell lines, *T. polium* methanolic extract showed a dose/time-dependent antitumor activity. The molecular docking approach revealed that most of the identified molecules were specifically binding with human peroxiredoxin 5, human androgen, and human progesterone receptors with high binding affinity scores. The obtained results confirmed that *T. polium* is a rich source of bioactive molecules with antioxidant and antitumor potential.

## 1. Introduction

Despite the arid and extra arid climate, the flora of Saudi Arabia is complex and contains more than 2285 species belonging to 855 genera [1,2]. In fact, 71.02% of these plants are herbs (1620 species), and many of them are classified as medicinal/aromatic ones. In the Hail region (northern central part of Saudi Arabia that extends between 250°29′ N and 380°42′ E Page: 2), the vegetation is influenced by those of the Mediterranean countries in the mountains and the Saharo-Arabian and Irano-Turranean phyto-geographical regions in An Nafud sand seas, open plains, and wadis [3]. A large majority of these plants have aromatic and medicinal virtues for the aromas they give off, their essential oil, and their rich content in polyphenols. In Saudi Arabia, more than 1200 (over 50%) of the total flowering plants (2250) are expected to be of medicinal importance.

The genus *Teucrium* includes more than 100 species and is largely distributed in Europe, North Africa, Asia, and especially in the Mediterranean region [4,5]. The Saudi Arabia flora comprises six *Teucrium* species: *T.* hijazicum Hedge & R.A. King, *T.* leucocladum Boiss, *T.* oliverianum Ging. exBenth, *T.* polium L., *T.* popovii R.A. King, and *T.* yemense Defl [6]. Most plant extracts and their bioactive molecules have been shown to be scavengers of free radicals, which form the basis of their therapeutic potential [7,8,9,10,11]. Their antioxidant nature has been closely linked with the cancer-preventing property of a plant-derived compound due to the fact that the inhibition of oxidative stress reduces mutations and chromosomal aberrations, which initiate carcinogenesis [12]. The antioxidant and anticancer properties of *T. polium* have been extensively studied over the years [13,14], which are attributed to certain identified polyphenolic compounds identified.

In fact, *Teucrium* members have been shown to contain different classes of compounds such as fatty acid esters, diterpenes, monoterpenes, sesquiterpenes, flavonoids, and polyphenolics [15,16,17]. Flavonoids that have been isolated from *T. polium* species include cirsimaritin, cirsiliol, cirsilineol, 5-hydroxy-6,7,30,40-tetramethoxyflavone, salvigenin, apigenin 5-galloylglucoside, apigenin-7-glucoside, vicenin-2-glucoside, and luteolin-7-glucoside [18,19,20,21]. In addition to their antioxidant activities, polyphenols have a wide range of biological activities [22,23,24,25,26]. Flavonoids also are known to protect the plant against ultraviolet radiation and possess anticancer [27,28], antioxidant, and anti-acetylcholinesterase activities [13,29,30,31,32]. It is also known that flavonoids possess antiviral, antifungal, and antibacterial properties [28,33]. To date, more than 134 bioactive compounds have been identified from different part of *T. polium* subspecies [34]. *Teucrium* species are used in folk medicine for treating many diseases such as abdominal pain, indigestion, common cold, diabetes, and urogenital diseases, and this plant has been reported to have hypolipidemic, antinociceptive, and anti-inflammatory effects [18,21]. It has been demonstrated that *T. polium* phytocompounds possess anti-diabetic, antiprofilative, pro-apoptotic, and anticancer activities [34,35,36,37,38].

The main objective of the present study was to investigate the phytochemical composition of *T. polium* methanolic extract using the HR-LCMS technique, the bioactive class of compounds (polyphenols, flavonoids, tannins…), and its antioxidant properties using four test systems. The antitumor effect was tested against two malignant cell lines: MatLyLu and Walker 256/B. The computational approach was used to confirm the antioxidant and anticancer activities of the identified compounds targeting the human peroxiredoxin 5 enzyme and human androgen/progesterone receptors.

## 2. Materials and Methods

### 2.1. Plant Material Sampling and Extract Preparation

The plant material was collected in October 2019 from a plant nursery in the Hail region (Saudi Arabia). The fresh aerial flowering parts (Figure 1) were dried at room temperature for ten days and then ground to a fine powder. Extracts were prepared according to Snoussi et al. [39]. Briefly, 40 g of the plant powder material were macerated in 400 mL of absolute methanol at room temperature for 48 h and re-extracted three times using the same procedure. Methanolic extracts were pooled, filtered, and the solvent was removed at 60 °C in the incubator chamber. The dried extracts were stored until further use. The yield was calculated using the following Formula (1):Yield (%) = (W1 × 100)/W2,(1)
where W1 was the weight of extract after the evaporation of solvent, and W2 was the dry weight of the sample.

### 2.2. Phytochemical Profile of T. polium Extract

#### 2.2.1. Phytochemical Analysis

The methanolic extract from the aerial part of the felty germander was qualitatively tested for the presence of alkaloids, flavonoids, terpenoids, tannins, saponins, steroids, proteins, amino acids, and cardiac glucosides by following the protocol described by Sofowora [40], Trease and Evans [41], and Adetuyi and Popoola [42].

#### 2.2.2. Identification of Bioactive by High Resolution-Liquid Chromatography Mass Spectroscopy

The phytochemical profile of the obtained crude methanolic extract from *T. polium* L. aerial parts was analyzed using a UHPLC-PDA-Detector 323 Mass Spectrophotometer (Agilent Technologies, Santa Clara, CA, USA). Compounds were identified via their mass spectra and their unique mass fragmentation patterns. Compound Discoverer 2.1, ChemSpider, and PubChem were used as the main tools for the identification of the phytochemical constituents [43].

### 2.3. Biological Activities

#### 2.3.1. Antioxidant Activities

##### DPPH Radical–Scavenging Activity

The ability to scavenge the 1,1-diphenyl-2-picrylhydrazyl (DPPH) radical was calculated using the following Formula (2) as described by Chakraborty and Paulraj [44]:DPPH scavenging activity (%) = (A_0_ − A_1_)/A_0_ × 100,(2)
where A_0_ is the absorbance of the control and A1 is the absorbance of the sample. The antioxidant activity was expressed as IC_50_ (mg/mL), which represented the extract concentrations scavenging 50% of DPPH radicals [45].

##### ABTS Radical Scavenging Activity Assay

The radical scavenging activity against ABTS (2,2′-azino-bis(3-ethylbenzothiazoline-6-sulfonic acid)) radical cations was measured using the method of Chakraborty and Paulraj [44]. The antiradical activity was expressed as IC_50_ (mg/mL), which represented the extract concentrations scavenging 50% of ABTS radicals [45]. The inhibition percentage of ABTS radical was calculated using the following Formula (3):ABTS scavenging activity (%) = (A_0_ − A_1_)/A_0_ × 100,(3)
where A_0_ is the absorbance of the control, and A1 is the absorbance of the sample.

##### Reducing Power Capability Assay

The reducing power was determined using the method of Bi et al. (2013). The extract concentration providing 0.5 of absorbance (IC_50_) was calculated from the graph of absorbance at 700 nm against sample concentration [46]. Ascorbic acid was used as a standard.

##### β-carotene/Linoleic Acid Method

The β-carotene method was carried out according to Ikram et al. [47]. Antioxidant activity (inhibition percentage, PI%) was evaluated using the following Formula (4):PI% = (A β-carotene T_120_/A β-carotene t_0_) × 100,(4)
where A β-carotene t_0_ and A β-carotene T_120_ refer to the corresponding absorbance values of the test sample standard and control measured before and after incubation for 2 h, respectively. All tests were performed in triplicate, and ascorbic acid (standard) was used for comparison.

### 2.4. In Vitro Anticancer Assessment Using Malignant MatLyLu and Walker 256/B Cell Lines and MTT Assay

Malignant MatLyLu (R33327) prostate cancer and Walker 256/B (W256) mammary gland cancer cells were used to test the in vitro anticancer activity of *T. polium* extract. These two cell lines were kindly provided by Prof. D. Chappard (Angers, France) to Dr. R. Badraoui (Sfax, Tunisia). MatLyLu and Walker 256/B malignant cells have a high osteolylic potential and are commonly used to induce osteosclerotic or osteolytic tumor lesions following the protocols previously described by Badraoui et al. [48,49].

MTT (3-[4จC-dimethylthiazole-2-yl]-2,5-diphenyltetrazolium bromide) assay was performed by the quantitative colorimetric method. Malignant Walker 256/B and MatLyLu (5 × 10^2^ cells/well) were seeded on 96-well plates with or without *T. polium* extract. The effect on the viability of the used cells was realized by using the following growing concentrations: 0–200 μg/mL. Pure ethanol was used as positive control. After 24 or 48 h, cells were incubated with MTT solution for 2 h. Then, the percentage of viability and inhibition was recorded by measuring the absorbance at 490 nm.

### 2.5. In Silico Study

The antioxidant activity of *T. polium* methanolic extract was confirmed by molecular docking of the identified phytochemical compounds from *T. polium* methanolic extract into the active site of the human peroxiredoxin 5 enzyme, the human progesterone (PR) enzyme, and human androgen receptor.

The intermolecular interactions between metabolites extracts and the active residues of peroxiredoxin 5 have been investigated using the AutoDock package [50]. The starting geometries of peroxiredoxin 5 and the original docked ligand benzoic acid were downloaded from the RCSB data bank web site: human peroxiredoxin 5 enzyme (PDB code 1HD2) [51], human progesterone (PDB code 4OAR) [46], human androgen receptor (PDB code 1E3G) [52].

The re-docking of the original ligand into the active site of the three tested target proteins are well reproduced with RMSD (root-mean-square deviation) values of 0.72, 1.14, and 0.651 Å, respectively, for peroxiredoxin 5 receptor, human progesterone receptor, and human androgen receptor. A stepwise molecular docking study was reported in previous study [48,53,54,55]. The docking calculations have been carried out using an Intel (R) Core (TM) i5-3770 CPU @ 3.40 GHz workstation.

### 2.6. Statistical Analysis

All measurements will be carried out in triplicate, and the results were presented as mean values ± SD (standard deviations).

## 3. Results

### 3.1. Phytochemical Composition

High Resolution-Liquid Chromatography Mass Spectrometry (HR-LCMS) was carried out the chemical composition of the active extract. This technique was performed in the separation and identification of the phytoconstituents based on their retention time, database difference (library), experimental *m/z*, MS/MS fragments, metabolite class, and proposed compounds. MS data were provided in negative and positive ionization mode. The majority of the *m/z* values in our extract were in the range from 133 to 742. In fact, HR-LCMS analysis identified peptide-like proteins in the methanolic extract of *T. polium*.

A total of 20 small peptides (tripeptides), with molecular weights ranging from 319 to 537 g/mol, were tentatively identified by comparison of spectrum data of the extract with that of known compounds. Details of identified peptides are given in Table 1.

Figure 2 summarizes the most dominant chemical compounds identified in *T. polium* methanolic extract by using HR-LCMS techniques.

Peptides of *T. polium* extract were composed of a majority of essential and non-essential amino acids distributed unevenly. Aromatic amino acids were predominant in 85% of the peptides with tryptophan (Trp) being the most commonly occurring amino acid (18.3%). Tryptophan along with phenylalanine and tyrosine accounted for 43.3% of the total amino acids, followed by glutamine, asparagine, and histidine (25%). Leucine, lysine, serine, glutamic acid, and cysteine together accounted for 16.7%; and proline and threonine made up 10% of the total amino acids. Arginine, isoleucine, and methionine were amongst the least abundant amino acids, which accounted for 5% of the amino acids in the identified peptides. Aromatic rings were the most abundant side chains followed by amide side chains. Sulfur-containing side chains were rare in the identified peptides.

The repertoire of peptides of *T. polium* extract appeared to be more hydrophobic than hydrophilic from the amino acid composition, with hydrophobic amino acids accounting for 51.7% of the total amino acids. In addition, the majority of the peptides were composed solely of hydrophobic amino acids. Neutral amino acids made up of 88% of the amino acids followed by basic amino acids (8%). Consequently, 14 of the 20 peptides were neutral in nature. Glutamic acid was the only acidic amino acid detected and was present in two of the 20 peptides identified. As seen in Table 2 (*m/z* values), most of the peptides had a net positive charge as determined by spectrophotometric data.

It is important to note that all compounds were first reported in this study for *T. polium* aerial parts methanolic extract analyzed with HR-LC/MS. The complete list of identified chemical bioactive compounds is summarized in Table 2.

### 3.2. Phytoconstituents and Antioxidant Activities

Phytochemical analysis showed the presence of diverse bioactive constituents such as saponins, cardiac glucosides, anthocyanin, terpenes, tannins, sterols, flavonols/flavanones, quinones, alkaloids, and coumarines. The results are summarized in Table 3.

The quantitative determination of phytochemical compounds specifies that the *T. polium* methanolic extract was rich in flavonoids (725 ± 0.001 mg QE/g extract), tannins (239 ± 0.006 mg QE/g extract), and phenols (72 ± 0.011 mg QE/g extract). The obtained *T. polium* methanolic extract was evaluated for its antioxidant potentiality using four methods. The IC50 of each test was calculated and determined (Table 4). As it is shown, this extract had the strong radical inhibition of ABTS (IC50 = 0.042 mg/mL) followed by DPPH (IC50 = 0.087 mg/mL), β-carotene/linoleic acid (IC50 = 0.101 mg/mL), and FRAP (IC50 = 0.292 mg/mL).

### 3.3. Anticancer Activities

The anticancer effect of the *T. polium* extract was investigated by MTT assay on two malignant lineages: mammary gland carcinoma (Walker 256/B cells) and prostate cancer (MatLyLu). Both cell lines have high metastatic potential and are commonly used to induce bone metastases [48,49]. The cells were treated with the plant extract at different concentrations 0, 50, 100, and 200 µg/mL for 24 or 48 h.

Our findings, validated by the antiproliferative effects on malignant Walker 256/B and MatLyLu cells, suggested that the methanolic extract of *T. polium* possess an anticancer effect. Its phytochemical profile might act as chemopreventive agents against both Walker 256/B and MatLyLu. In fact, the extract suppressed the growth of the two malignant lines once compared with the controls (0 µg/mL) (Figure 3). The lowest viability was noticed with 200 µg/mL of *T. polium* extract for both cell lines. Overall, the effect was accentuated with the dose increase (dose-dependent), and it was more prominent after 48 h of treatment; the effect was dose and time-dependent.

### 3.4. In Silico Study

Overall, the in silico approach showed that for antioxidant and antitumor tests, the activity differs from one compound to another. These differences can be explained by the structural geometry of its basic skeleton, and to the presence of different specific substituted groups and heteroatoms in the studied bioactive compounds. In an attempt to rationalize the observed antioxidant activity of the identified molecules in *T. polium* extract, a molecular docking study has been carried out to determine their binding modes from one side and from another site of the active residues of human peroxiredoxin 5. The results of the number of conventional intermolecular hydrogen bonding established between the docked compounds and active site residues of human peroxiredoxin 5 are summarized in Table 5.

All the complexes formed between the composition of *T. polium* methanolic extract (Table 5) and the active residues of Peroxiredoxin 5 have negative binding energies, which may explain the potent antioxidant activity of *T. polium* methanolic extract. The band energies of the stable complexes range from −8.06 to −2.01 kcal mol^−1^. According to the molecular docking results, the amino sugar 13 showed the lowest binding energy (−8.06 kcal mol^−1^), and thus, the highest antioxidant activity was well fitted into the binding cavity of human peroxiredoxin 5. This amino sugar forms five strong hydrogen bonding with amino acids ALA A42, THR A44, THR A147, GLY A46, and CYS A47 at distances of 2.04, 2.88, 2.54, 3.11, and 2.81Å, respectively (Figure 4), as well as a carbon hydrogen bond with PRO A45.

It appears from the docking outputs in Table 5 that the antioxidant activity varies with subclass family. For instance, lemonoids 14 (−7.09 kcal mol^−1^) and 28 (−6.14 kcal mol^−1^) showed higher binding affinity compared with other subclasses (Table 5).

Table 6 summarized the calculated binding energies of the stable complexes ligand– progesterone, the number of conventional intermolecular hydrogen bonding established between the docked compounds, and the active site residues of progesterone.

All the complexes formed between the composition of *T. polium* methanolic extract (Table 6) and the active residues of progesterone showed negative binding energies, which may explain the observed anticancer activity of *T. polium* methanolic extract. The bond energies of the stable complexes range from −10.83 to −3.61 kcal mol^−1^. According to molecular docking results, limonoids 28 and 14 showed the higher anticancer activity with lowest binding energies of −10.83 and −10.45 kcal mol^−1^. The higher activity of 28 compared with 14 may be due to the extra intermolecular types of hydrogen bonding, and π-sulfur types appear in the former compared with the latter (Figure 5). Indeed, in the stable complex 28-progesterone, two strong hydrogen bonds were formed between 28 and the progesterone receptor. The first one is formed between the lone pair of oxygen atoms of the furan ring and the amino acid GLN A725 at a distance of 3.15 Å, and the second one at a distance of 3.06 Å is established between the lone pair of an oxygen atom of cyclopentanone and VAL A760. π-Sulfor intermolecular interactions are formed between the π bonds of furan ring and the amino acids MET A759 and MET A801 (Figure 5) of distances 5.44 and 5.29 Å, respectively.

Table 7 summarizes the calculated binding energies of the stable ligand–progesterone complexes, the number of conventional intermolecular hydrogen bonds established between the docked compounds, and the active site residues of the human androgen receptor. The complexes formed between the composition of *T. polium* methanolic extract (Table 7) and the active residues of androgen showed negative and positive binding energies. On one hand, the complexes that show negative binding energies may explain the anticancer activity of *T. polium* methanolic extract. On the other hand, the positive binding energies may indicate that the corresponding metabolites are not active; i.e., they have no anticancer activity. The bond energies of the stable complexes range −11.01 to −3.49 kcal mol^−1^. According to the binding energies, compound 18 showed the higher anticancer activity with the lowest binding energy of 11.01 kcal mol^−1^ (Table 7 and Figure 6).

The stability of the 18-androgen complex may refer to the strong interactions formed between 18 and amino acids of androgen (Figure 5). Indeed, two strong hydrogen bonds were formed between the acetyl groups of 18 and amino acids THR A877 and ARG 752 of the androgen receptor at distances of 3.32 and 3.40 Å, respectively (Figure 6). Furthermore, a strong sulfur-X bond is formed between the oxygen atom of the acetyl group of 18 and the sulfur atom of the methylthiol moiety of MET A780 at a distance of 3.17 Å (Figure 6).

## 4. Discussion

### 4.1. Phytochemical Composition of T. polium Extract

According to our result, terpenoids, also known as isoprenoids, are the most abundant compound class in the methanolic extract of *T. polium* aerial parts as defined by the HR-LCMS technique. Among the identified compounds (Table 1), many secondary metabolites with known antioxidants and anticancer activities belong to different classes of bioactive molecules, including isoprenoid, fatty acids, amino fatty acids, amino alcohols, glycerolipids, amino sugars, phenol, alkaloids, flavanol, small peptides, etc. [55,56]. It has been demonstrated that Liquid Chromatography Mass Spectrometry is a very sensitive method, which can identify many new compounds. Indeed, Aghakhani et al. [57] isolated twenty-two new flavonoid compounds that were first reported for *Phlomis* species. Our results are in agreement with previous studies that indicated that the *Teucrium* genus contains different classes of phytoconstituents such as monoterpenes, sesquiterpenes [58], polyphenols, flavonoids [59,60], and fatty acid esters [61,62]. Moreover, flavonoids are polyphenols that are detected in medicinal plants with a wide variety of biological activities [63]. Many previous studies have described the presence of various flavonoids such as apigenin, luteolin, rutin, cirsiliol, cirsimaritin, salvigenin, and eupatorin in the roots, aerial parts, and inflorescences of the plant [17,59,60,61,62,63,64,65,66]. Furthermore, one intermediate in the biosynthetic pathways of alkaloids, iridoid glycoside (Harpagoside), was also detected [60]. This compound has been detected in a wide variety of plants and in some animals. Furthermore, two iridoid glycosides, teucardoside and teuhircoside, from the hydrophilic fraction of *T. polium var. pilosum* and *T. polium var. Alba* were isolated [34]. Elmasri et al. [37] isolated iridoid and phenylethanol glycosides and a monoterpenoid from the areal part of *T. polium*. It was reported that 3′,4′,5trihydroxy-6,7-dimethoxy-flavone exhibited an inhibition of the biofilm-forming strain *Staphylococcus aureus* [37]. It was reported by Sharififar et al. [65] that the methanolic extract of the areal part of *T. polium* contains four major flavonoids where rutin and apigenin were found to be the most active fractions as radical scavengers. In 2018, Özer et al. [66] used the LC-MS/MS method to analyze compounds and to investigate the antioxidant activity of *T. polium* L. in decoction and infusion. Among the secondary metabolites, polyphenols represent an interesting class that have biological activities such as antioxidant and anticancer [67,68]. Moreover, flavonoids, another secondary metabolites class, are widely produced by plants to fight against biotic and abiotic aggression. It has been well demonstrated that this compounds exhibited antioxidant and anticancer activities [69,70]. The degree of polymerization and the differences arising from hydroxyl group substitutions makes this class of metabolites very large, with about 4000 different compounds [69].

Additionally, during our investigation, we identified 20 small peptide fragments with varying amino acid sequences via HR-LCMS analysis. Therefore, it was of interest to analyze the role of these peptides in the antioxidant and anticancer properties of the plant extracts. Peptides and peptide-like proteins have been found to be integral components of various plant species and have therapeutic applications due to their broad spectrum of biological activities such as antimicrobial, antioxidation, antihypertensive, immunomodulatory, and anticancer properties [71]. Although peptides from different plant species vary greatly with respect to amino acid composition and sequence, bioactive peptides possess certain common features such as small size, hydrophobicity, large percentage of aromatic amino acids, and amphiphatic nature [72].

### 4.2. Antioxidant Activities

The antioxidants of *T. polium* extracts have been widely studied [73,74,75]. For instance, using DPPH and β-carotene/linoleic acid assays, respectively, the petroleum ether (IC50 = 73.2; 9.2 µg/mL), chloroform (IC50 = 85.4; 5.1 µg/mL), methanol (IC50 = 20.1; 25.8 µg/mL), and water (IC50 = 40.6; 19.2 µg/mL) extracts of this plant which were collected from Kerman (Iran) showed strong antioxidant activity compared to our findings [65]. The aqueous extract of *T. polium*, collected from Israel, was evaluated for its antioxidant potentiality using various tests [76]. It inhibited superoxide radical (IC50 = 12.0 µg/mL), hydroxyl radical (IC50 = 66.0 µg/mL), β-carotene (inhibition percentage IP = 60% at 100µg/mL), iron-induced lipid peroxidation (IC50 = 7.0µg/mL), 2,20-azobis (2-amidinopropan) dihydrochloride (AAPH)-induced plasma oxidation (IP = 84% at 667µg/mL). It had also the capacity to bind iron (IC50 = 79.0 µg/mL). At 1 mg/mL, this aqueous extract had the ability to increase intracellular GSH (glutathione) levels in cultured HepG2 cells [76]. The 80% ethanolic extract of *T. polium*, collected from Iran, was determined for its in vivo antiradical activity using 1,1-diphenyl-2-picrylhydrazyl (DPPH) radical (78.6 and 90.7%), total antioxidant power (23.6 and 37.5%), and thiobarbituric acid reactive substances (24.7 and 31.8%) in serum at 50 and 100 mg/kg, respectively [77].

Our study showed the strong antioxidant capacity of the methanolic extract. This activity could not be related to the total phenolic, (0.072 mg GAE/g extract), flavonoid (0.725 mg QE/g extract), or tannin contents (0,239 mg TAE/g extract), and the extract was analyzed by HR-LCMS in order to identify the major active compounds (Table 1). The results showed the presence of various chemical compounds, such as fatty acids, terpenes, alkaloids, coumarines, and flavonoids. These classes are already known by their antioxidant abilities [78,79], and thus, they may explain the present activity. The analyzed peptides showed in Table 2 can also explain in part the antioxidant activity [80,81,82].

Furthermore, a previous study showed that the isoprenoid 10-hydroxyloganin had an insecticidal effect [83]. The limonoid khayanthone was detected in *Punica granatum* methanolic extract, which exhibited antioxidant potentiality [84]. The compounds 13R-hydroxy-9E,11Z-octadecadienoic, acid bis (2-hydroxypropyl) amine, 9-aminononanoic acid, and 10-aminodecanoic acid found in this plant, which belong to fatty acids, amino alcohols, and amino acid derivatives, were proved in the literature to have antiradical activity [85,86]. Cepharanthine is widely known in several clinical uses. It was used for the treatment of various diseases such as the inhibition of free radicals, radiation-induced leukopenia, venomous snakebites, etc. [87,88]. At 30 μg/mL, cepharanthine had the ability of 94.6% inhibition on the peroxidation of linoleic acid [89]. It also exhibited activity using DPPH, ABTS, superoxide anion, ferrous ion chelating, total antioxidant activity, hydrogen peroxide, reducing power, and N,N-dimethyl-p-phenylenediamine dihydrochloride radicals scavenging [89]. Moreover, harpagoside inhibited free radicals at 1 mg/mL [90].

The relationship between the structure and activity of natural peptides with antioxidant properties has been vividly elucidated in a report by Zou et al. [91] where certain aspects of the chemical structure, namely the small size, presence of certain amino acids in large amounts, and hydrophobicity, are described as important factors that influence the antioxidant nature of peptides. Similar findings have been reported in several plant peptides such as those from *Sphenostylis stenocarpa* [92], hemp seed [93], phaseolin, bean [94], and *Jatropha curcas* [95] using several in vitro antioxidant evaluation systems such as diphenyl-1-picryhydradzyl (DPPH) and linoleic acid oxidation. In our study, many of the above-mentioned characteristics (small size, hydrophobicity, and high occurrence of aromatic amino acids) were seen in the peptides from *T. polium*, which further confirms the potential of the plant extract as a therapeutic agent.

### 4.3. Anticancer Activities of T. polium Extract

The anticancer effect of the Teucrium polium extract was investigated by MTT assay on two malignant lineages: mammary gland carcinoma (Walker 256/B cells) and prostate cancer (MatLyLu). Both cell lines have high metastatic potential and are commonly used to induce bone metastases [42,43]. The cells were treated with the plant extract at different concentrations 0, 50, 100 and 200 µg/mL for 24 or 48 h.

As shown in Figure 3, *T. polium* extract seems to possess an anticancer effect. In fact, it inhibits the proliferation of both malignant Walker 256/B and MatLyLu cells in a concentration- and time-dependent manner. It has been previously reported that *T. polium* inhibited the proliferation of prostate cancer cells [96]. Similarly, several medicinal plant extracts inhibit the invasion, cancer evolution, and metastases. A recent study expected that in the near future, *T. polium* extract may be a novel anticancer agent [97]. This possibility is certainly related to the promising effects of the plant extract and could explain the ethno-pharmacological applications and the traditional use of *T. polium*. In fact, as shown in Table 1 and Table 2, the plant methanolic extract exhibits a relevant and promising phytochemical composition following HR-LCMS assay. It includes, but is not limited to, flavonoid and isoflavonoid, isoprenoid, diterpene triepoxide (such as triptonide), and terpene (such as valtratum). All these chemical compounds possess pharmacological activities. Basically, these natural compounds, such as the triptonide, were effective in inhibiting tumorigenicity and tumor growth in a wide variety of cancers, including pancreatic cancer and thyroid induced metastases by activating the tumor-suppressive MAPK (mitogen-activated protein kinase) signaling pathway and via astrocyte elevated gene-1, respectively [98,99]. These phytochemical compounds might have better pharmacological properties together rather than separated. In fact, it has been reported that the effect of the whole plant is usually much better than that of its active phytochemical compounds [100].

Previous studies with *T. polium* extract reported potential anticancer effects by the inhibition of cell invasion and motility of human prostate cancer [96]. The mechanism includes E-caderin/catenin complex restoration. In this study, an anticancer effect has been revealed on the prostate MatLyLu cell line. Similarly, an in vivo study of prostate cancer and its lymph node, lung, and bone metastases complication due to MatLyLu cells could inhibit the invasion and metastatic potential via E-caderin/catenin complex restoration.

The effect of related plants showed efficient effects against breast and prostate cancer. In fact, extract from *T.* persicum was reported to inhibit PC-3 prostate cancer cells proliferation [101]. It has been demonstrated that both *T.* capitatum and *T.* creticum were effective against MCF-7 breast cancer cells [102]. Moreover, *T.* romasissimun and has an anticancer potential by inhibiting K562 proliferation [103]. Nevertheless, the reported IC50 values were different, and that could be related to the area from which the plant has been collected and/or the cancer cell types. However, it is well known that *T.* polium extract may also potentiate the apoptotic effects of some anticancer drugs such as doxorubicin and vinblastine on several cancer cell lines, including Saos-2, A431, SW480, and Skmel-3 [104]. Further preclinical investigations concerning the in vivo anticancer effects on Walker 256/B, MatLyLu, and other cancer types might be of potential interests to confirm the protective effect of *T.* polium and its possibility to contribute to the discovery of new drug products to treat cancer.

### 4.4. In Silico Study

Our molecular docking results with antioxidant human peroxiredoxin 5 were perfectly correlated with those of Eze et al. [105], confirming the same and highest binding pose of 1-(phenylsulphonyl)-N-propylpyrrolidine-2-carboxamide in the binding cavity of human peroxiredoxin 5 with a binding energy of (−13.86 kcal mol^−1^) and interactions with THR44, PRO40, PRO 45, GLY46, ARG127, THR147, and CYS A47 residues. Similarly, α-tocopherol (−7.2 kcal mol^−1^) *Cymbopogon citratus* essential oil was perfectly fitted into the cavity of peroxiredoxin 5 establishing H-bonding with Arg 127(A) and non-covalent interactions with ASP A113, THR A147, LEU A116, SER A115, LEU A112, PRO A40, THR A44, GLY A46, CYS A47, PHE A120, and ASP A145. Additionally, caryophyllene oxide (−7.1 kcal mol^−1^) from the same essential oil was interacting with non-covalent interactions with PRO A40, THR A147, THR A44, PHE A120, PRO A45, LEU A116, and ILE A119 with amino of peroxiredoxin 5 amino acids residues, confirming therefore the high potency *T. polium* methanolic extract.

The obtained molecular docking interactions (Figure 5) of *T. polium* methanolic extract into the active site of the human progesterone enzyme were in good agreement with those reported by Acharya et al. [46]. In fact, these authors demonstrated that furanocoumarins, xanthotoxol, bergapten, angelicin, psoralen, and imperatonin are potent anti-breast cancer agents against progesterone. These molecules were buried into the active site of the human progesterone enzyme via the following molecular interactions: LEU718, GLN725, MET 759, LEU763, ARG766, PHE778 (xanthotoxol), GLN725, LEU721, LEU718, MET756, MET759, LEU763, ARG766, PHE778, MET801(bergapten), LEU718, LEU721, GLY722, GLN725, MET759, LEU763, LEU763, ARG766, PHE778 (angelicin), LEU718, LEU721, GLN725, MET756, MET759, LEU763, ARG766, PHE778, MET801 (psoralen) and LEU718, LEU721, GLY722, GLN725, MET756, LEU763, ARG768, PHE778, LEU797, MET801, and LEU887 (imperatonin).

Our results indicated that our selected compounds displaying the lowest binding energies share a higher number of common residues with the active sites of receptor 1E3G having the corresponding interacting residues, LEU 701, LEU 704, ASN 705, LEU 707, GLY 708, GLN 711, TRP 741, MET742, MET 745, VAL 746, MET 749, ARG 752, PHE 764, MET 780, MET787, LEU 873, PHE 876, THR 877, LEU 880, MET 895, and ILE 899. The docking results were well supported by those in vitro showing that *T. polium* methanolic extract can be a new potential resource of natural antioxidant and anticancer compounds.

## 5. Conclusions

The overall hydrophobic nature of peptides identified in *T. polium* methanolic extracts coupled with the small size of peptides and high concentration of aromatic amino acids ascertain its antioxidant and anticancer characteristics. Further in vivo studies are necessary to confirm the pharmacological properties of the identified molecules in vivo as promising antioxidant and antitumor agents.

## Figures and Tables

**Figure 1 antioxidants-09-01089-f001:**
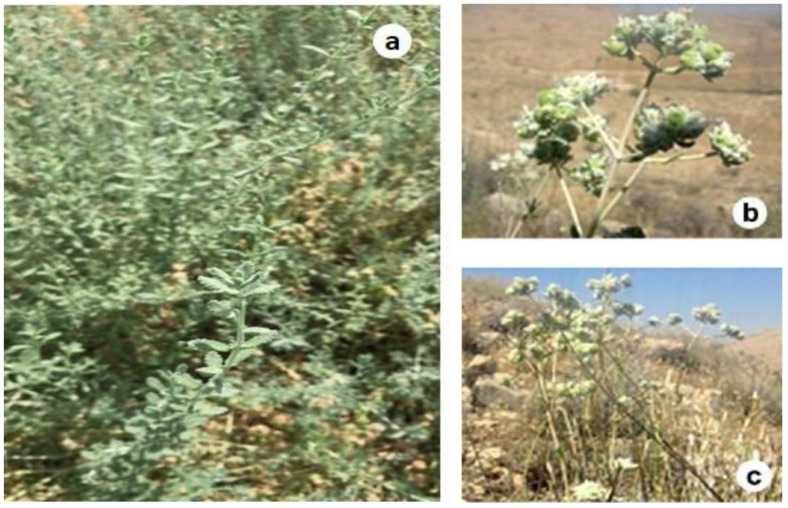
Vegetative growth (**a**), budding (**b**), and full bloom (**c**) of *T. polium* collected from Hail region.

**Figure 2 antioxidants-09-01089-f002:**
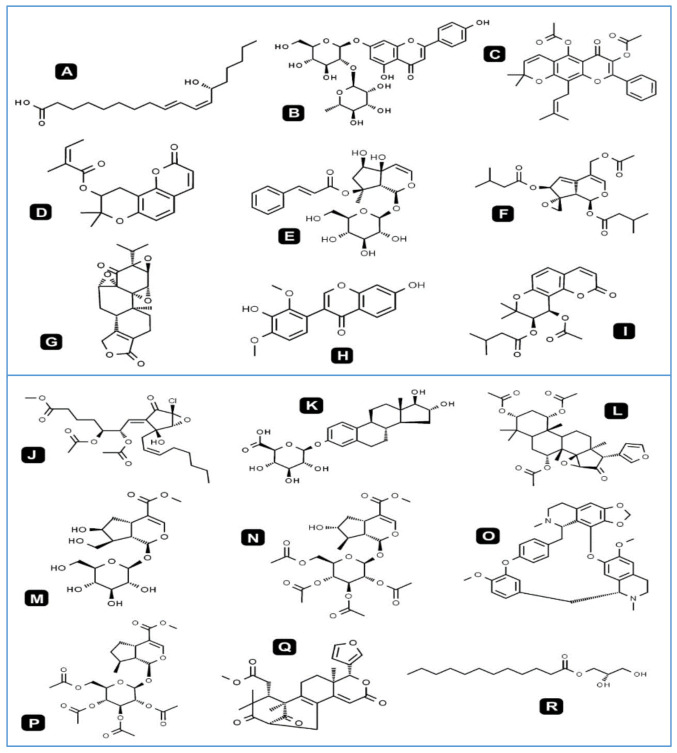
Chemical structures of the most dominant identified compounds in *T. polium* methanolic extract by using High Resolution-Liquid Chromatography Mass Spectrometry (HR-LCMS). (**A**). 13R-hydroxy-9E,11Z-octadecadienoic acid, (**B**). Rhoifolin, (**C**). Sericetin diacetate, (**D**). Selinidin, (**E**). Harpagoside, (**F**). Valtratum, (**G**). Triptonide, (**H**). Koparin 2′-Methyl Ether, (**I**). Dihydrosamidin (**J**). 10S,11R-Epoxy-punaglandin, (**K**). 4, 16alpha, 17beta-Estriol 3-(beta-D-glucuronide), (**L**). Khayanthone. (**M**). 10-Hydroxyloganin, (**N**). 7-Epiloganin tetraacetate, (**O**). Cepharanthine, (**P**). Deoxyloganin tetraacetate, (**Q**). Carapin-8 (9)-Ene, (**R**). 1-dodecanoyl-sn-glycerol.

**Figure 3 antioxidants-09-01089-f003:**
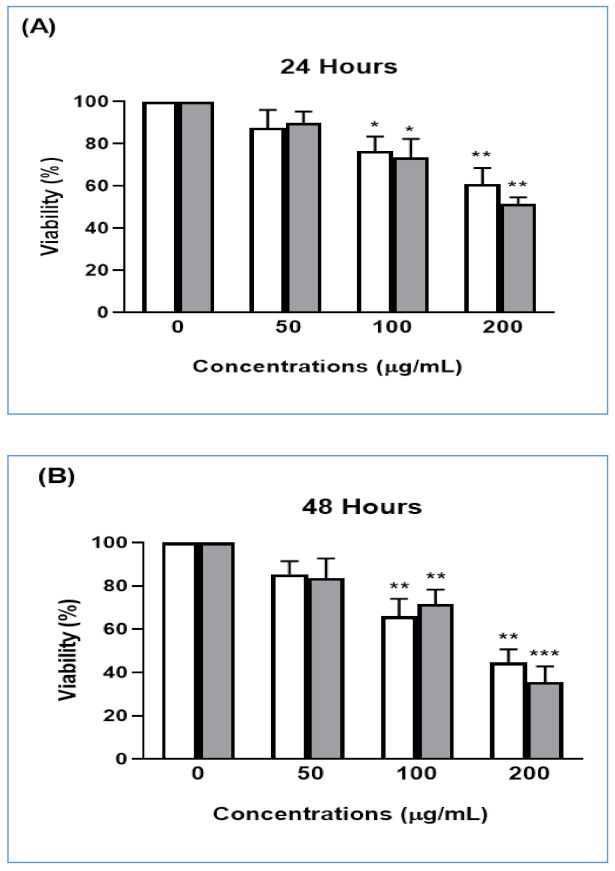
Anticancer effect of *T. polium* methanolic extract on malignant Walker 256/B mammary gland carcinoma cells (White) and MatLyLu prostate cancer cells (gray) in 24 (**A**) and 48 h (**B**). Legend: * *p* < 0.05, ** *p* < 0.01, *** *p* < 0.001.

**Figure 4 antioxidants-09-01089-f004:**
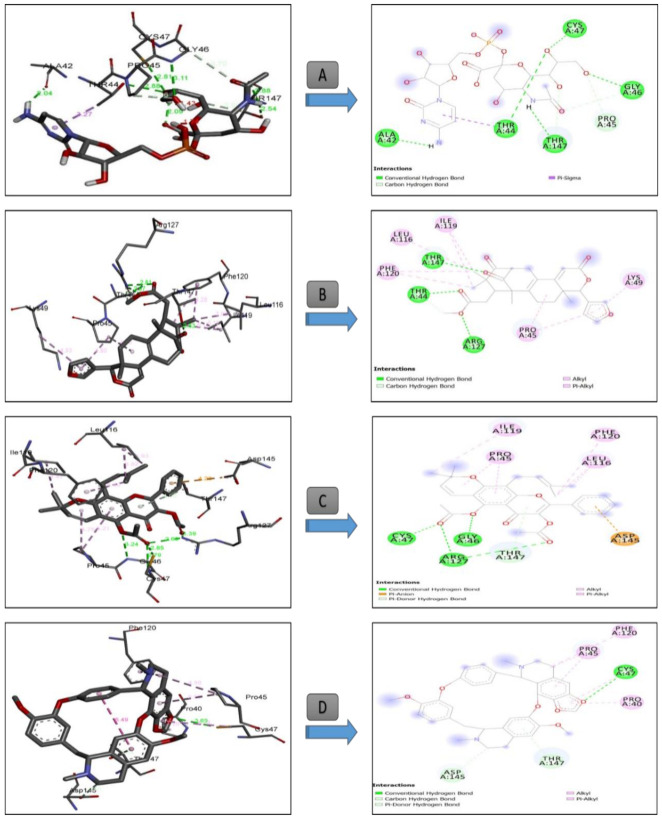
Three-dimensional (**right**) and two-dimensional (**left**) closest interactions between active site residues of peroxiredoxin 5 and some molecules belonging to different class compounds with the best score result. Legend: (**A**): Compound **13** (CMP-N-acetylneuraminic acid, Class: Amino Sugar), (**B**): Compound **14** (Carapin-8 (9)-Ene, Class: Limnoid), (**C**): Compound **10** (Sericetin diacetate, Class: Flavonol), and (**D**): Compound **8** (Cepharantine, Class: Alkaloid).

**Figure 5 antioxidants-09-01089-f005:**
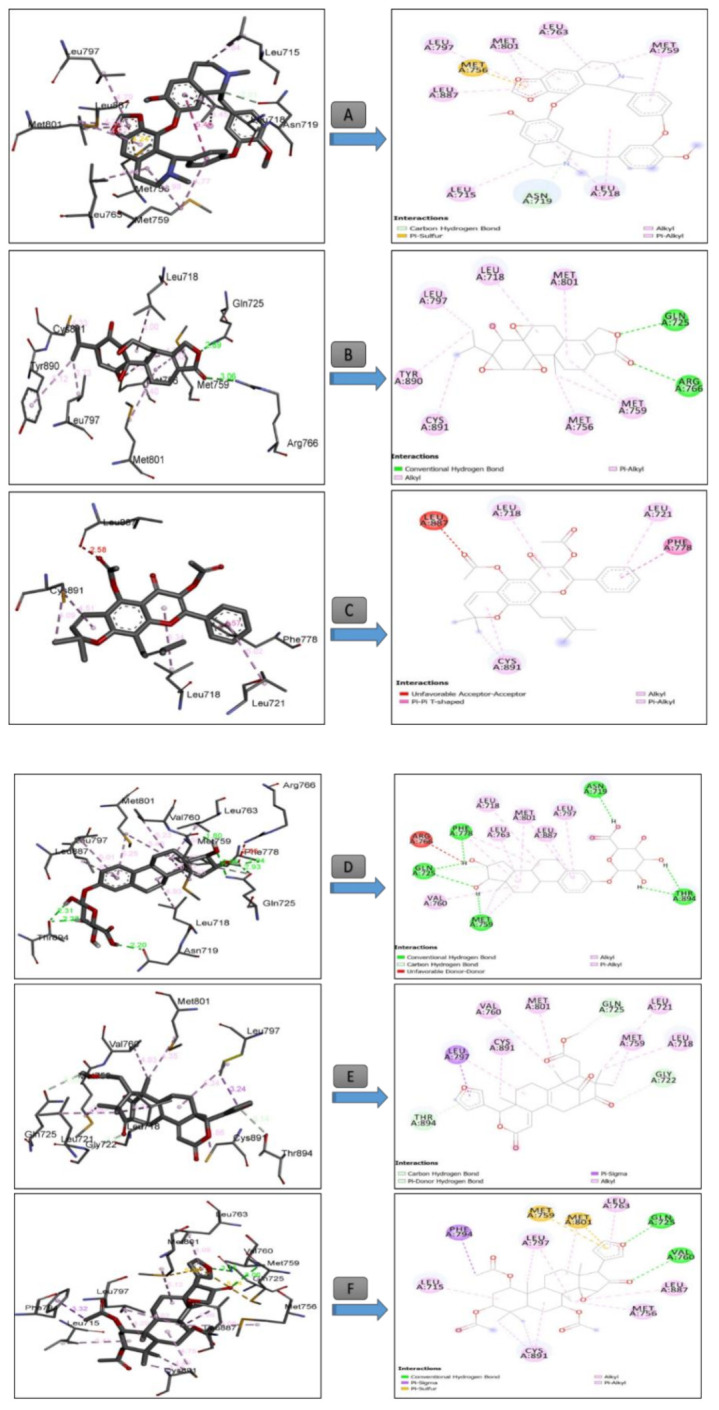
Three-dimensional (**right**) and two-dimensional (**left**) closest interactions between active site residues of the human progesterone receptor and some molecules belonging to different class compounds with the best score result. Legend: (**A**): Compound **8** (Cepharantine, Class: Alkaloid), (**B**): Compound **20** (Triptonide, Class: Diterpene triepoxide), (**C**): Compound **10** (Sericetin diacetate, Class: Flavonol), (**D**): Compound **24** (16alpha,17beta-Estriol 3-(beta-D-glucuronide, Class: Steroidal glycosides), (**E**): Compound **14** (Carapin-8 (9)-Ene, Class: Limnoid), (**F**): Compound **28** (Khayanthone, Class: Limnoid).

**Figure 6 antioxidants-09-01089-f006:**
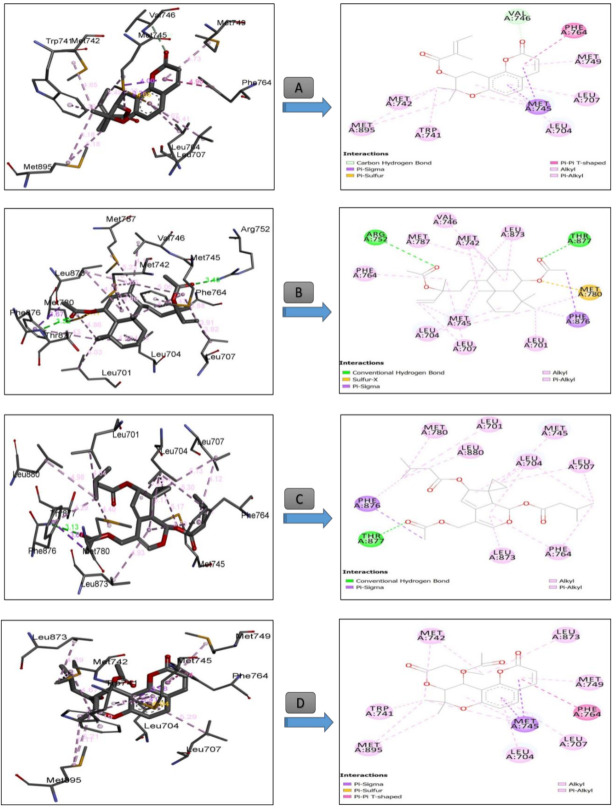
Three-dimensional (**right**) and 2D **(left**) closest interactions between the active site residues of the human androgen and some molecules belonging to different class compounds with the best score result. Legend: (**A**): Compound **15** (Selinidin, Class: coumarin derivative), (**B**): Compound **18** (larixol acetate), (**C**): Compound **19** (Valtratum, Class: Terpene) and (**D**): Compound **22** (dihydrosamidin, Class: coumarins).

**Table 1 antioxidants-09-01089-t001:** Peptide-like proteins identified by the HR-LCMS technique in *T. polium* methanolic extract.

Small Peptides	Retention Time (mn)	Molecular Weight	Formula	[*m/z*]−	[*m/z*]+
Asn Asn Asn	0.945	360.1384	C_12_H_20_N_6_O_7_	341.1201	-
His Phe Gln	3.988	430.1976	C_20_H_26_N_6_O_5_	411.1797	-
Gln His Phe	4.05	430.1978	C_20_H_26_N_6_O_5_	447.1566	-
Thr Leu Trp	6.593	418.2222	C_21_H_30_N_4_O_5_	435.1808	-
Arg Glu Trp	7.029	489.2233	C_22_H_31_N_7_O_6_	-	512.2123
Gln Phe Tyr	7.241	456.202	C_23_H_28_N_4_O_6_	491.1714	-
Trp Phe Trp	7.837	537.2444	C_31_H_31_N_5_O_4_	-	373.1268
Phe Tyr Gln	8.417	456.2011	C_23_H_28_N_4_O_6_	-	183.1149
Gln Phe Phe	8.42	440.2065	C_23_H_28_N_4_O_5_	-	174.1475
Gln Tyr Trp	9.304	495.2108	C_25_H_29_N_5_O_6_	-	341.1374
Thr Leu Ser	9.441	319.1731	C_13_H_25_N_3_O_6_	-	342.1705
Tyr Glu Trp	9.572	496.1978	C_25_H_28_N_4_O_7_	-	95.0482
Pro Trp Pro	9.663	398.1964	C_21_H_26_N_4_O_4_	-	95.0479
Trp Tyr Gln	9.76	495.2113	C_25_H_29_N_5_O_6_	-	95.0486
Asn His Met	9.986	400.1527	C_15_H_24_N_6_O_5_S	-	95.0490
Thr Trp Phe	10.036	452.2084	C_24_H_28_N_4_O_5_	-	95.0499
Lys His Cys	10.229	386.1734	C_15_H_26_N_6_O_4_S	-	149.0968
Lys Phe Cys	10.598	396.1822	C_18_H_28_N_4_O_4_S	-	95.0491
Trp Ser Tyr	10.78	454.187	C_23_H_26_N_4_O_6_	-	95.0489
Trp Pro Ile	12.677	414.2271	C_22_H_30_N_4_O_4_	-	299.1620

**Table 2 antioxidants-09-01089-t002:** Phytochemical composition of *T. polium* methanolic extract using the HR-LCMS technique.

N°	Identified Compound Name	Class of Compounds	RT [min]	Formula	[M + H]^+^ (*m/z*)	[M + H]^−^ (*m/z*)
**1**	10-Hydroxyloganin	Isoprenoid	0.945	C_17_H_26_O_11_	-	406.1439
**2**	13R-Hydroxy-9E,11Z octadecadienoic acid	Octadecanoid	1.046	C_18_H_32_O_3_	296.232	-
**3**	Bis (2-hydroxypropyl) amine	Amino Alcohol	1.062	C_6_H_15_NO_2_	133.1111	-
**4**	9-Aminononanoic acid	Amino Fatty Acid	1.447	C_9_H_19_NO_2_	173.1401	-
**5**	10-Aminodecanoic acid	Amino Fatty Acid		C_10_H_21_NO_2_	187.156	-
**6**	7-Epiloganin tetraacetate	Isoprenoid	4.739	C_25_H_34_O_14_	-	558.1987
**7**	b-D-Glucopyranoside uronic acid, 6-(3-oxobutyl)-2- naphthalenyl	Organic Acid, Phenol	5.342	C_20_H_22_O_8_	-	390.1284
**8**	Cepharanthine	Alkaloid	5.948	C_37_H_38_N_2_O_6_		606.2582
**9**	Rhoifolin	Flavonoid	6.376	C_27_H_30_O_14_	578.1634	-
**10**	Sericetin diacetate	Flavonol	7.838	C_29_H_28_O_7_	488.191	-
**11**	Troxerutin	Flavonol	5.96	C_33_H_42_O_19_	-	742.2379
**12**	Deoxyloganin tetraacetate	Isoprenoid	6.319	C_25_H_34_O_13_	-	542.2048
**13**	CMP-N-acetylneuraminic acid	Amino Sugar	6.329	C_20_H_31_N_4_ O_16_P	-	614.1563
**14**	Carapin-8 (9)-Ene	Limonoid	8.198	C_27_H_30_O_7_	-	466.1996
**15**	Selinidin	Coumarin Derivative	8.848	C_19_H_20_O_5_	328.1303	-
**16**	Harpagoside	Iridoid Glycoside	9.1	C_24_H_30_O_11_	494.1782	-
**17**	8-Epiiridodial glucoside tetraacetate	Isoprenoid	9.126	C_24_H_34_O_11_	498.2126	-
**18**	Larixol Acetate	-	9.262	C_22_H_36_O_3_	348.2635	-
**19**	Valtratum	Terpene	9.525	C_22_H_30_O_8_	422.1935	-
**20**	Triptonide	Diterpene triepoxide	9.807	C_20_H_22_O_6_	358.1419	-
**21**	Koparin 2′-Methyl Ether	Isoflavonoid	10.036	C_17_H_14_O_6_	314.0792	-
**22**	Dihydrosamidin	Coumarins	10.779	C_21_H_24_O_7_	388.1524	-
**23**	10S,11R-Epoxy-punaglandin 4	Eicosanoid	11.022	C_25_H_35_ClO_9_	514.1895	-
**24**	16Alpha,17beta-Estriol 3-(beta-D-glucuronide)	Steroidal glycosides	11.279	C_24_H_32_O_9_	464.2051	-
**25**	16-Hydroxy-4-carboxyretinoic Acid	Isoprenoid	11.28	C_20_H_24_O_5_	344.1621	-
**26**	Isotectorigenin, 7- Methyl ether	IsoFlavonoid	12.149	C_18_H_16_O_6_	328.0939	-
**27**	3-hydroxy-3′,4′- Dimethoxyflavone	Flavonoid	13.274	C_17_H_14_O_5_	298.0829	-
**28**	Khayanthone	Limonoid	18.427	C_32_H_42_O_9_	570.2854	-
**29**	1-Dodecanoyl-sn-glycerol	Glycerolipid	20.37	C_14_H_22_N_2_O_3_	-	266.1651

**Table 3 antioxidants-09-01089-t003:** Qualitative analysis of phytochemicals in methanolic extract of *T. polium* aerial parts.

Phytochemical Compounds	SAP	CAG	ANT	TER	FLAV	TAN	ST	QUI	COU	FLA	FAT	ALK
*T. polium* extract	(++)	(++)	(+)	(++)	(+)	(+)	(++)	(+)	(+)	(+)	(+)	(+)

SAP: saponins, CAG: cardiac glucosides, ANT: anthocyanin, TER: terpenes, FLAV: flavonols and flavanones, TAN: tannins, ST: sterols, QUI: quinones, COU: coumarines, FL: flavonoids, FAT: fatty acids, ALK: alkaloids. (+): presence, (-): absent, (++): abundant.

**Table 4 antioxidants-09-01089-t004:** Antioxidant activities of *T. polium* + methanolic extract as compared to ascorbic acid and BHT.

Test Systems	*T. polium* Methanolic Extract	(BHT)	(AA)
**Phytochemical Composition**			
**Total Flavonoids Content (mg QE/g Extract)**	725 ± 0.001	-	-
**Total Tannins Content (mg TAE/g Extract)**	239 ± 0.006	-	-
**Total Phenols Content (mg GAE/g Extract)**	72 ± 0.011	-	-
**Antioxidant Activities**			
**DPPH IC_50_ (mg/mL)**	0.087 ± 0.001 ^b^	0.023 ± 3 × 10^−4 a^	0.022 ± 5 × 10^−4 a^
**ABTS IC_50_ (mg/mL)**	0.042 ± 0.014 ^b^	0.018 ± 4 × 10^−4 a^	0.021 ± 1 × 10^−3 a^
**β-carotene IC_50_ (mg/mL)**	0.101 ± 0.020 ^c^	0.042 ± 3.5 × 10^−3 b^	0.017 ± 1 × 10^−3^
**FRAP IC_50_ (mg/mL)**	0.292 ± 0.042 ^c^	0.05 ± 0.003 ^a^	0.09 ± 0.007 ^b^

BHT: butylated hydroxytoluene, AA: ascorbic acid. The letters (a–c) indicate a significant difference between the different antioxidant methods according to the Duncan test (*p* < 0.05).

**Table 5 antioxidants-09-01089-t005:** Docking binding energies, conventional hydrogen bonding, and the number of closest residues to the docked compounds into the active site of human peroxiredoxin 5.

Compound No.	Class of Compounds	Free Binding Energy (kcal/mol)	Conventional H-Bonds (HBs)	Number of Closest Residues to the Docked Ligand in the Active Site
**1**	Isoprenoid	−3.82	6	7
**2**	Octadecanoid	−5.35	6	5
**3**	Amino Alcohol	−2.15	3	2
**4**	Amino Fatty Acid	−4.54	6	6
**5**	Amino Fatty Acid	−4.60	5	5
**6**	Isoprenoid	−4.66	5	6
**7**	Organic Acid, Phenol	−5.05	3	6
**8**	Alkaloid	−5.07	1	6
**9**	Flavonoid	−5.09	8	11
**10**	Flavonol	−6.00	4	9
**11**	Flavonol	−2.01	8	9
**12**	Isoprenoid	−5.13	5	8
**13**	Amino Sugar	−8.06	5	6
**14**	Limonoid	−7.09	3	8
**15**	Coumarin Derivative	−5.94	3	7
**16**	Iridoid Glycoside	−4.78	6	6
**17**	Isoprenoid	−5.25	5	6
**18**	-	−6.03	2	7
**19**	Terpene	−5.17	5	8
**20**	diterpene triepoxide	−5.70	5	5
**21**	Isoflavonoid	−4.70	5	6
**22**	Coumarins	−5.37	4	6
**23**	Eicosanoid	−3.18	2	6
**24**	Steroidal Glycosides	−5.45	5	6
**25**	Isoprenoid	−5.15	3	7
**26**	IsoFlavonoid	-	-	-
**27**	Flavonoid	−5.11	2	7
**28**	Limonoid	−6.14	4	8
**29**	Glycerolipid	−2.68	6	5

**Table 6 antioxidants-09-01089-t006:** Docking binding energies, conventional hydrogen bonding, and the number of closest residues to the docked compounds into the active site of the human progesterone.

No.	Class of Compounds	Free Binding Energy (kcal/mol)	Conventional H-Bonds (HBs)	Number of Closest Residues to the Docked Ligand in the Active Site
**1**	Isoprenoid	−7.10	8	8
**2**	Octadecanoid	−6.84	3	4
**3**	Amino Alcohol	−3.61	3	4
**4**	Amino Fatty Acid	−3.69	5	3
**5**	Amino Fatty Acid	−4.78	4	4
**6**	Isoprenoid	−7.83	3	7
**7**	Organic Acid, Phenol	−7.61	4	8
**8**	Alkaloid	−8.56	0	9
**9**	Flavonoid	−8.46	4	12
**10**	Flavonol	−9.48	0	5
**11**	Flavonol	−6.41	7	10
**12**	Isoprenoid	−7.97	3	9
**13**	Amino Sugar	−3.90	4	9
**14**	Limonoid	−10.45	0	10
**15**	Coumarin Derivative	−8.38	1	8
**16**	Iridoid Glycoside	−7.41	5	10
**17**	Isoprenoid	−8.07	2	10
**18**	-	−9.01	2	9
**19**	Terpene	−8.13	3	8
**20**	Diterpene triepoxide	−9.28	2	9
**21**	Isoflavonoid	−7.70	3	8
**22**	Coumarins	−8.78	2	12
**23**	Eicosanoid	−6.89	2	5
**24**	Steroidal Glycosides	−10.06	7	12
**25**	Isoprenoid	−8.63	4	11
**26**	IsoFlavonoid	−6.69	2	8
**27**	Flavonoid	−7.03	2	9
**28**	Limonoid	−10.83	2	11
**29**	Glycerolipid	−5.56	3	2

**Table 7 antioxidants-09-01089-t007:** Docking binding energies, conventional hydrogen bonding, and the number of closest residues to the docked compounds into the active site of the human androgen receptor.

No.	Class of Compounds	Free Binding Energy (kcal/mol)	Conventional H-Bonds (HBs)	Number of Closest Residues to the Docked Ligand in the Active Site
**1**	Isoprenoid	−7.61	5	7
**2**	Octadecanoid	−8.01	3	4
**3**	Amino Alcohol	−3.51	3	3
**4**	Amino Fatty Acid	−5.74	5	3
**5**	Amino Fatty Acid	−6.13	5	3
**6**	Isoprenoid	−7.18	6	8
**7**	Organic Acid, Phenol	−8.31	2	7
**8**	Alkaloid	+134.65	0	5
**9**	Flavonoid	+19.33	1	10
**10**	Flavonol	−3.49	0	11
**11**	Flavonol	+29.13	4	11
**12**	Isoprenoid	−5.95	0	9
**13**	Amino Sugar	+2.64	4	8
**14**	Limonoid	−4.94	3	13
**15**	Coumarin Derivative	−9.70	0	9
**16**	Iridoid Glycoside	−8.07	3	9
**17**	Isoprenoid	−7.73	2	14
**18**	-	−11.01	2	13
**19**	Terpene	−9.66	1	10
**20**	Diterpene Triepoxide	−9.61	1	14
**21**	Isoflavonoid	−8.96	4	9
**22**	Coumarins	−9.81	0	9
**23**	Eicosanoid	−4.42	2	7
**24**	Steroidal Glycosides	+17.12	4	11
**25**	Isoprenoid	−4.96	3	12
**26**	IsoFlavonoid	−8.29	3	9
**27**	Flavonoid	−8.37	1	7
**28**	Limonoid	+9.39	0	7
**29**	Glycerolipid	−5.89	2	2

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
