# Peer review of "HR-LCMS-Based Metabolite Profiling, Antioxidant, and Anticancer Properties of Teucrium polium L. Methanolic Extract: Computational and In Vitro Study"

_antioxidants, 2020, doi:10.3390/antiox9111089_

Round 1

Reviewer 1 Report

"HR-LCMS based metabolite profiling, Antioxidant, and Anticancer Properties of Teucrium polium L. methanolic extract: Computational and in vitro Study" is a well-articulated research manuscript with international collaborations which has investigated the phytochemical profile, anticancer and antioxidant activities of Teucrium polium methanolic extract using both in vitro and in silico approaches.

The results obtained clearly indicate that T. polium is a rich source of bioactive molecules with antioxidant activity and could be used as a safe antitumor drug.

Comments:

  1. Research with natural compounds is generally complex and the authors need to show is there are potential toxicity issues with this compound.
  2. The authors have used only one type of in vitro assay that is MTT I would recommend the use of a second assay such as MTS assay to further validate and strengthen their results. 
  3. There are some typographical errors that need to be corrected and the manuscript needs to be proofread by a native English speaker. 

Author Response

Point-by-point response to Reviewer 1:

Dear Dr.  (Reviewer 1): Thank you for your consideration of our manuscript. We found the comments helpful, and believe our revised manuscript represents a significant improvement over our initial submission.

"HR-LCMS based metabolite profiling, Antioxidant, and Anticancer Properties of Teucrium polium L. methanolic extract: Computational and in vitro Study" is a well-articulated research manuscript with international collaborations which has investigated the phytochemical profile, anticancer and antioxidant activities of Teucrium polium methanolic extract using both in vitro and in silico approaches.

The results obtained clearly indicate that T. polium is a rich source of bioactive molecules with antioxidant activity and could be used as a safe antitumor drug.

Comments:

  1. Research with natural compounds is generally complex and the authors need to show is there are potential toxicity issues with this compound.

In fact, some previous studies reported the cytotoxic power of T. polium plant extract.

Nematollahi-Mahani et al. [51] studied the toxicity of T. polium 96%-ethanolic extract on different known malignant cell lines like A549 (human lung adenocarcinoma), BT20 (human breast ductal carcinoma), MCF-7 (human breast adenocarcinoma), and PC12 (mouse pheochromocytoma). The results reported showed that IC50 values were 90 µg/ml (A549), 106 µg/ml (BT20), 140 µg/ml (MCF-7) and 120 µg/ml (PC12). The cytotoxic properties of T. polium aerial parts can be attributed the presence in the organic extracts of some diterpenoids and their acyl derivatives (teucvin and teucvidin), flavonoids (cirsiliol, cirsimaritin, cirsilineol, salvigenin and 5-hydroxy-6,7,3’,4’-tetramethoxyllavone), saponin poliusaposide [52], and selenium [53-55].

  1. Nematollahi-Mahani, N.; Rezazadeh-Kermani, M.; Mehrabani, M.; Nakhaee, N. Cytotoxic Effects of Teucrium polium on Some Established Cell Lines. Pharm. Biol. 2007, 45, 295–298.
  2. Elmasri, A.; Hegazy, M.E.F.; Mechrefa, Y.; Paré, W.P. Cytotoxic saponin poliusaposide from Teucrium polium. R.S.C. 2015, 5, 27126–27133.
  3. Nagao, ; Ito, N.; Kohno, T.; Kuroda, H.; Fujita, E. Antitumor activity of Rabdosia and Teucrium diterpenoids against P388 lymphocytic leukemia in mice. Chem. Pharm. Bull. (Tokyo) 1982, 30, 727–729.
  4. Mseddi, ; Alimi, F.; Noumi, N.; Deshpande, S.; Adnan, M.; Hamdi, A.; Elkahoui, S.; Alghamdi, A.; Kadri, A.; Patel, P.; et al. Thymus musilii Velen. as a promising source of potent bioactive compounds with its pharmacological properties: In Vitro and in silico analysis. Arab. J. Chem. 2020, 13, 6782–6801.
  5. Jurisic, V.K.S.; Kalodera, Z.; Grgic, J. Determination of selenium in Teucrium generation atomic absorption spectrometry. Z. Nat. 2003, 58, 143–145.
  1. The authors have used only one type of in vitro assay that is MTT I would recommend the use of a second assay such as MTS assay to further validate and strengthen their results. 

The anti-tumoral effect of T. polium extract has been checked in vitro using PBS containing Trypan Blue exclusion assay but the results are included in another study performed in the same laboratory. The effect was assessed and compared with a bisphosphonate and the results are mirroring the T. polium extract which is reported in the current study.

  1. There are some typographical errors that need to be corrected and the manuscript needs to be proofread by a native English speaker. 

Paper was deeply rechecked for English style and grammatical errors by all authors and an English  native speaker.

Reviewer 2 Report

I have submitted a revised manuscript with some grammar changes.  Congratulations for doing some much work to support your conclusions.

Author Response

Point-by-point response to Reviewer 2:

Dear Dr.  (Reviewer 2): Thank you for your consideration of our manuscript. We found the comments helpful, and believe our revised manuscript represents a significant improvement over our initial submission.

Open Review

English language and style

( ) Extensive editing of English language and style required
(x) Moderate English changes required
( ) English language and style are fine/minor spell check required
( ) I don't feel qualified to judge about the English language and style

Yes

Can be improved

Must be improved

Not applicable

Does the introduction provide sufficient background and include all relevant references?

(x)

( )

( )

( )

Is the research design appropriate?

(x)

( )

( )

( )

Are the methods adequately described?

(x)

( )

( )

( )

Are the results clearly presented?

(x)

( )

( )

( )

Are the conclusions supported by the results?

(x)

( )

( )

( )

Comments and Suggestions for Authors

I have submitted a revised manuscript with some grammar changes.  Congratulations for doing some much work to support your conclusions.

Dear colleague. Special thanks for your comments. We updated our manuscript according to your comments.

For the controls used in anticancer activity: Cell lines in culture medium containing all additives except plant extract (0 mg/ml).

Submission Date

25 September 2020

Date of this review

16 Oct 2020 00:59:19

Round 2

Reviewer 1 Report

I am satisfied with the revisions, and now the paper is accepted for publication 

Author Response

Comments and Suggestions for Authors

I am satisfied with the revisions, and now the paper is accepted for publication 

Dear colleague

Special thanks for your comments who made our paper better

All the best